# The First Polish Isolate of a Novel Species *Pectobacterium aquaticum* Originates from a Pomeranian Lake

**DOI:** 10.3390/ijerph18095041

**Published:** 2021-05-10

**Authors:** Weronika Babinska, Agata Motyka-Pomagruk, Wojciech Sledz, Agnieszka Kowalczyk, Zbigniew Kaczynski, Ewa Lojkowska

**Affiliations:** 1Laboratory of Plant Protection and Biotechnology, Intercollegiate Faculty of Biotechnology, University of Gdansk, 58 Abrahama, 80-307 Gdansk, Poland; weronika.babinska@phdstud.ug.edu.pl (W.B.); agata.motyka-pomagruk@ug.edu.pl (A.M.-P.); wojciech.sledz@biotech.ug.edu.pl (W.S.); 2Laboratory of Structural Biochemistry, Faculty of Chemistry, University of Gdansk, 63 Wita Stwosza, 80-308 Gdansk, Poland; agnieszka.kowalczyk@ug.edu.pl (A.K.); zbigniew.kaczynski@ug.edu.pl (Z.K.)

**Keywords:** *Pectobacteriaceae*, pectinolytic bacteria, soft rot, blackleg, monitoring, natural waterways

## Abstract

Pectinolytic bacteria from the genus *Pectobacterium* cause high economic losses in various crops, vegetables, and ornamentals including potato. Thus far, these strains have been isolated from distinct environments such as rotten or asymptomatic plants, soil, and waterways. The prevalence of soft rot *Pectobacteriaceae* in different depths of Pomeranian lakes was performed by a qualified scuba diver over 2 years of monitoring. It allowed for the isolation and broad characterization of a strain from the newly established species *Pectobacterium aquaticum*. Phylogenetic analysis on the sequences of *dnaX* and *recA* genes revealed the highest similarity of this strain to *P. aquaticum* CFBP 8637^T^. In addition to the determination of analytical profile index (API 20E), we discovered that this strain possesses a smooth form of a lipopolysaccharide with O-polysaccharide consisting of mannose, glucose, and abequose. Moreover, the characterized strain, described as *P. aquaticum* IFB5637, produced plant-cell–wall-degrading enzymes, such as pectinases, cellulases, proteases, and was capable of macerating potato and chicory tissues under laboratory conditions. In view of more frequent irrigation of seed potato fields resulting from the ongoing climate warming, it is important to monitor the occurrence of potential disease-causing agents in natural waterways.

## 1. Introduction

Bacteria classified to the *Pectobacterium* genus are Gram-negative, non-sporulating, facultative anaerobes [1]. Members of this genus joined the newly established *Pectobacteriaceae* family [2] due to the fact of their genomic characteristics and their ability to produce pectinolytic enzymes. These bacteria cause disease symptoms of blackleg on potato plants and soft rot on potato tubers and other vegetables, e.g., tomatoes, peppers, chicory, and ornamental plants [3,4,5]. The virulence of these pathogens results from the production of large quantities of plant-cell-wall-degrading enzymes (PCWDEs), i.e., pectinases, cellulases, and proteases that are secreted outside bacterial cells and allow for efficient maceration of plant tissue [6]. Apart from the above-listed PCWDEs, pectinolytic bacteria possess an effective iron chelating system that enables iron accumulation and survival in spite of the limited availability of this element [7]. In addition, motility turned out to be highly relevant for the pathogenesis of *Pectobacterium* spp. [8,9], as by these means bacteria are able to enter the plant through wounds or natural openings, which leads to successful colonization and maceration of the host tissue.

It needs to be underlined that bacteria belonging to the genus *Pectobacterium* are highly heterogeneous [10,11,12,13] and differ in their ability to cause disease symptoms on diverse plant species [14] under varying temperature conditions [15]. For example, *Pectobacterium atrosepticum* is responsible for disease outbreaks only under temperate climate [1,12], while *Pectobacterium brasiliense* is capable of triggering symptomatic infections in tropical and subtropical regions [16,17]. On the other hand, *Pectobacterium parmentieri* (until 2016 *Pectobacterium wasabiae*) was isolated from the affected plants in many European countries with various climate conditions (e.g., Norway, Poland, and Spain), in addition to such distant geographical areas such as New Zealand, South Africa, and Canada [10,11,13,18,19,20]. Referring to the broad host range of *Pectobacterium* spp., *Pectobacterium carotovorum* causes disease symptoms on potato, other vegetables, and ornamentals [21,22], while, for instance, *P. brasiliense* was observed to trigger soft rot in *Capsicum annum* L., *Ornithogalum* spp. and *Daucus carota* subsp. *sativus* [23].

The high significance of *Pectobacterium* spp. is underlined by enclosing these microbes in the top 10 list of bacterial plant pathogens based on scientific/economic importance [24]. Especially, notable annual economic damage results from the activity of *Pectobacteriaceae* in the potato production sector [1]. From an epidemiological point of view, it is extremely important that in spite of high economic losses and broad host range associated with pectinolytic bacteria, there is a lack of effective plant protection strategies against these phytopathogens [25].

Taking into account that *Pectobacterium* spp. spread by the latently infected seed tubers, they are classified as seed borne pathogens. In addition, soil, water, plant remains, insects, and nematodes, and agricultural machines are considered as important sources of pectinolytic bacteria [25,26,27,28,29]. Moreover, excessive irrigation and mechanical damage during harvest or insufficient ventilation in storage aggravate the severity of the resultant disease symptoms [4]. There were several reports suggesting that surface waters may be a potential sources of *Pectobacterium* spp. [30]. These communications considered the presence of *P. carotovorum* and *P. atrosepticum* in waters originating from drains, ditches, streams, rivers, and lakes in the USA and Scotland [31,32,33]. Moreover, *Pectobacterium versatile* was detected in waterways such as an alpine river stream, a river crossing arable land, and downstream from an irrigation canal [34]. Recently, a novel bacterial species, namely *Pectobacterium aquaticum* has been established [35]. By now, isolates from this taxonomic group have been isolated solely from three different water reservoirs in southeastern France (2015–2016) [35]. The abovementioned reports confirm that surface waters, which may potentially be used for irrigation purposes, are likely to harbor pectinolytic bacteria.

Our research group has been monitoring the occurrence of pectinolytic bacteria on seed potato plantations in Poland since 1996 [5]. The members of *Pectobacteriaceae* were also detected in some of the tested waterways in our country [36]. Recently, we undertook monitoring of the prevalence of pectinolytic bacteria at different depths in nine Pomeranian lakes. On this basis, we describe the first isolation, identification, and characterization of *P. aquaticum* strain originating from a natural water reservoir in Poland.

## 2. Materials and Methods

### 2.1. Water Samples Collection

Water samples were collected from the following 9 lakes located in the Pomeranian region in northern Poland: Wiejskie (54°01′14′ N 17°17′13′ E) (54°02′10.6″ N 17°28′02.9″ E), Czarne Dabrowno (54°07′44.1′ N 17°36′08.2′ E) (54°13′18.48″ N 17°60′58.18″ E), Wdzydze (53°58′31′ N 17°54′19′ E) (53°57′18.1″ N 17°54′19.9″ E), Biale (54°22′32′ N 18°11′13′ E) (51°49′80.7″ N 23°53′56.5″ E), Radunskie (54°16′11′ N 18°01′16′ E) (54°16′72.2″ N 18°16′66.8″ E), Klodno (54°19′02′ N 18°06′32′ E) (54°32′05.44″ N 18°10′99.80″ E), Jelen (54°12′04.4′ N 17°31′31.5′ E) (54°12′04.4″ N 17°31′31.5″ E), Grabowskie (54°06′15′ N 18°09′02′ E) (54°15′44.79″ N 18°15′58.54″ E), and Grabowko (54°15′09.47′ N 18°18′69.6′ E) (54°15′09.47″ N 18°18′69.6″ E). In the years 2016 and 2017, 50 mL samples were gathered from the investigated reservoirs by a qualified scuba diver. The samples were taken from the nearby shore as well as from the depths of 0 m, 5 m, 10 m, 15 m, and 20 m. In addition, one slime sample was also acquired at the depth of 20 m. The samples were collected in such a way as to avoid mixing of waters originating from different depths. Due to the isolation of pectinolytic bacteria from Jelen Lake, the procedure for water sample collection from this reservoir was repeated every two months. Finally, this lake was sampled 6 times.

### 2.2. Isolation and Identification of Pectinolytic Strains

One hundred microliters of each water sample were plated on a semi-selective crystal violet pectate (CVP) medium [37] and incubated at 28 °C for 48 h. Bacterial colonies forming characteristic cavities were picked up and repeatedly plated in a reductive manner on CVP and tryptone soya agar (TSA) (Oxoid, Basingstoke, UK) as many times as it was necessary to reach the axenic culture state. All axenic pectinolytic isolates were subsequently frozen in 40% glycerol and kept at −80 °C for further analyses.

In order to assign pectinolytic isolates to the proper taxonomic groups, the bacterial biomass of the acquired strain was collected from the frozen stock and streaked in a reductive manner on TSA. After 24 h incubation at 28 °C, a single bacterial colony was suspended in 200 µL of sterile distilled water and frozen for 45 min at −20 °C and then defrosted. This procedure allows for effective lysis of the bacterial cells. Two microliters of such lysate were subsequently subjected to PCR-based identification.

At first, it was investigated whether the stated bacterial isolate belonged to the highly heterogeneous *P. carotovorum* group. This assignment was conducted by a multiplex PCR assay [38]. This reaction was carried out in a 25 μL PCR mixture containing 2 μL of the tested bacterial lysate, 1× reaction buffer supplemented with KCl (Thermo Fisher Scientific, Minneapolis, MN, USA), 2.5 mM MgCl_2_, 80 μM of each dNTPs, 0.32 μM Df and Dr primers (for *Dickeya* genus), 0.1 μM Y45 and Y46 primers (for *P. atrosepticum*), 1.2 μM ExpccF and ExpccR primers (for bacteria formerly classified as *P. carotovorum*) in addition to 1 U of the *Taq* recombinant DNA Polymerase (Thermo Fisher Scientific, Minneapolis, MN, USA). The amplification was carried out with the use of the Thermal Cycler C1000 75 (Bio-Rad, Hercules, CA, USA) according to the following procedure: initial denaturation (95 °C, 4 min), 30 cycles of denaturation (94 °C, 45 s), annealing (62 °C, 90 s), and extension (72 °C, 90 s), with a final single extension step (72 °C, 3 min). Electrophoretic separation of the resulting amplicons was conducted in 1.5% agarose gel in 0.5× tris–borate–EDTA buffer at 100 V for 40 min. A 100 bp DNA molecular ladder (Thermo Fisher Scientific, Minneapolis, MN, USA) was used for the assessment of the length of the resultant PCR amplicons. After, 5 mg mL^−1^ ethidium bromide was utilized for staining. Then, the visualization process was performed under UV light in ChemiDoc XRS system (Bio-Rad, Hercules, CA, USA). The isolates that gave specific PCR product of 535 bp with ExpccF and ExpccR primers were classified to the *P. carotovorum* species, and *P. carotovorum* SCRI 136 was used as a reference strain (Table 1).

Afterwards, we evaluated whether a strain classified to the *P. carotovorum* group should be assigned to *P. brasiliense* (formerly *P. carotovorum* subsp. *brasiliense*) species. Therefore, a single PCR assay was applied as described by Duarte et al. [16]. This PCR reaction was also performed in a total volume of 25 μL consisting of 2 µL of bacterial lysate and 1× PCR buffer supplemented with KCl, 2.5 mM MgCl_2_, 0.1 mM of dNTPs, 1 µM BR1f and L1r primers, 1 U *Taq* recombinant DNA Polymerase. DNA amplification was performed under the following conditions: initial denaturation (94 °C, 2 min), 25 cycles of denaturation (94 °C, 45 s), annealing (62 °C, 45 s), and extension (72 °C, 90 s) with a final single extension step (72 °C, 10 min). Electrophoretic separation was carried out as mentioned above.

The further to-species identification of the acquired pectinolytic isolate was based on phylogenetic analysis and genomic profiling as described below.

### 2.3. Phylogeny and Genomic Profiling-Based Identification of IFB5637

Genomic DNA of the obtained pectinolytic isolate and reference strains (Table 1) was extracted from an overnight culture in tryptone soya broth (TSB) (BTL, Łodz,.Poland) of the stated strain using a commercially available Genomic Mini AX Bacterial Kit (A&A Biotechnology, Łodz,.Poland). The quality and concentration of the isolated DNA was assessed by using a NanoDrop ND-1000 (Thermo Fisher Scientific, Minneapolis, MN, USA).

The reassignment of the acquired pectinolytic isolate from *P. brasiliense* to *P. aquaticum* species was based on the comparison of the sequences of *dnaX* [45] and *recA* [46] genes. The isolated DNA of *P. aquaticum* IFB5637 and the reference strains (Table 1) was diluted to the concentration of 10 ng µL^−1^. [36]. In the case of *dnaX,* PCR was performed in a 50 µL final solution mixture containing 2 µL of the previously diluted bacterial DNA, 1× PCR buffer supplemented with KCl, 2.5 mM MgCl_2_, 0.2 mM of each dNTPs, 0.4 μM of dnaXF and dnaXR primers, and 1 U *Taq* DNA Polymerase (Thermo Fisher Scientific, Minneapolis, MN, USA). The amplification was conducted using the following protocol: initial denaturation (94 °C, 3 min), 35 cycles of denaturation (94 °C, 1 min), annealing (59 °C, 1 min), and extension (72 °C, 2 min) with a final single extension step (72 °C, 5 min). In the case of *recA,* PCR was performed in the 25 µL final solution mixture containing 1 µL of the previously diluted bacterial DNA, 1× PCR buffer supplemented with KCl, 0.25 mM MgCl_2_, 0.13 mM of each dNTPs, 1 μM of recA1 and recA2 primers, and 1 U *Taq* recombinant DNA Polymerase. The amplification was conducted using the following protocol: initial denaturation (95 °C, 5 min), 32 cycles of denaturation (94 °C, 1 min), annealing (47 °C, 1 min), and extension (72 °C, 2 min) with a final single extension step (72 °C, 5 min).

The resultant PCR products of 535 bp for *dnaX* and 730 bp for *recA* were sequenced from both ends by a commercial company (Genomed, Warsaw, Poland). The chromatographic data were manually edited and aligned using the CLC Main Workbench version 6.9.1 (CLC Inc., Aarhus, Denmark) with default parameters. The edited sequences of *dnaX* and *recA* genes of the isolated strain were compared to *dnaX* and *recA* gene sequences available in the GenBank database. Concerning *dnaX* based phylogenetic tree reposted in this work, the following sequence was used GCA_003382645.2 for CFBP 8632; GCA_003382585.2 for CFBP 8635; GCA_003382645.2 for CFBP 8633; GCA_003382655.2 for CFBP 8636; GCA_003382595.2 for CFBP 8634 and TMK516879.1 for CFBP 8637TS. In addition, the sequences for *P. brasiliense* CFBP 6617 (MK516956.1), *P. carotovorum* (MW657239 for IFB5639; MK516909.1 for CFBP 2046), *P. atrosepticum* (BX950851.1 for SCRI 1046; MK516904.1 for CFBP 1526), *P. parmentieri* (CP003415.1 for SCC3193; MK516972.1 for CFBP 8475), and *D. dadantii* CFBP1269 (JX434940.1) were used for constructing a phylogenetic tree in MEGA X (Pennsylvania State University; www.megasoftware.net (accessed on 9 May 2021)) [47]. The tree was generated with a neighbor-joining algorithm with the Jukes–Cantor nucleotide distance measure. Bootstrap was set on the level of 1000 replicates.

Genomic profiling of the acquired pectinolytic isolate, named IFB5637, and the *P. aqauticum* reference strains (Table 1) were based on repetitive sequence PCR (rep-PCR). Rep-PCR reaction was conducted with ERIC primers according to Versalovic et al. [48]. The isolated DNA of *P. aquaticum* IFB5637 *P. aquaticum* and reference strains (Table 1) was diluted to the concentration of 10 ng µL ^−1^ [36]. PCR was performed in a 25 µL final solution containing 5 µL of the diluted bacterial DNA, 1 × PCR buffer supplemented with (NH_4_)_2_SO_4_, 3.5 mM MgCl_2_, 0.1 mM of each dNTPs, 0.4 μM of ERIC1F and ERIC2R primers, and 2 U *Taq* recombinant DNA Polymerase. Amplification was conducted using the following protocol: initial denaturation (95 °C, 7 min), 30 cycles of denaturation (94 °C, 1 min), annealing (53 °C, 1 min), and extension (65 °C, 8 min) with a final single extension step (65 °C, 16 min). Electrophoretic separation of the resulting amplicons was conducted in 1% agarose gel in 0.5 × tris–borate–EDTA buffer at 50 V for 4 h. The outcomes were visualized in the same way as in the case of single PCRs as described above. A 1 kb DNA molecular ladder was used (Thermo Fisher Scientific, Minneapolis, MN, USA).

### 2.4. Phenotypic Characterization of the Collected IFB5637

Overnight bacterial cultures on TSA plates were prepared from the frozen socks of the collected IFB5637 isolate and the reference strains (Table 1). One bacterial colony was picked from each TSA plate and used for inoculation of the 5 mL TSB medium. Twenty-four hour incubation at 28 °C with 120 rpm shaking followed. These cultures were subsequently centrifuged (6000 rpm for 10 min), and the bacterial pellets were washed twice in 0.85% NaCl to remove any media residuals. Bacterial suspensions of OD_600_ = 0.1 (approximately 10^8^ CFU mL^−1^) in 0.85% NaCl solution were prepared for IFB5637 and the reference strains (Table 1) and used in the following procedures if not stated otherwise.

#### 2.4.1. Pathogenicity Assays

The ability of IFB5637 isolate to cause disease symptoms was evaluated on potato slices and chicory leaves as described previously by Zoledowska et al. [11] and Van Gijsegem et al. [49], respectively.

Regarding the potato slices maceration assay, potato tubers cv. Lord were washed under tap water and sterilized with 10% sodium hypochlorite solution for 20 min. Later, the tubers were air-dried under laminar flow cabinet and then aseptically cut into 10 mm thin slices. Subsequently, in each slice 5 mm diameter holes were drilled. Each hole in every potato slice was inoculated with 50 µL of OD_600_ = 0.1 bacterial suspension. The slices were placed on wet linen in plastic boxes covered with lids for assuring high relative humidity and incubated at 28 °C for 48 h. Diameters of the rotten spots were measured afterwards. As a negative control, 0.85% NaCl was used. Three biological repetitions of this experiment, each with nine technical replications, were carried out.

Concerning pathogenicity on chicory leaves, sterile pipette tips were used for drilling leaves at the base of each chicory leaf. The leaves were inoculated with 10 µL of OD_600_ = 0.1 bacterial suspension and incubated in plastic bags at 28 °C for 48 h in high relative humidity. Subsequently, the length of the rotten tissue was measured. Three biological repetitions of the experiment, involving ten chicory leaves per strain, were performed.

#### 2.4.2. Activity of Plant-Cell-Wall-Degrading Enzymes

Bacterial ability to produce plant-cell-wall-degrading enzymes, including pectinases, cellulases, lipases, and proteases, was assessed according to the previously described protocols [50,51,52,53].

The activity of pectinases was tested on a M63 + 0.25% poligalacturonate (PGA) medium [50]. Two microliters of the OD_600_ = 0.1 bacterial suspension were spotted on this plate and incubated at 28 °C for 48 h. An aqueous 10% copper acetate solution was subsequently applied on the surface of the medium for 10 min to visualize PGA degradation zones. The resultant halo zones around bacterial colonies were measured.

The activity of cellulases was evaluated on a M63 + 1% carboxymethylcellulose (CMC) medium [51]. Two microliters of the OD_600_ = 0.1 bacterial suspension were spotted on this plate and incubated at 28 °C for 48 h. Afterwards, the medium was stained with 1% Congo red dye solution for 5 min and washed two times with 4 M NaCl solution. Brown halo zones visible around the colonies, which corresponded to the exhibited cellulase activity, were measured.

Lipase activity was analysed with the use of a Rhodamine medium [52]. Two microliters of the OD_600_ = 0.1 bacterial suspension were spotted on this plate and incubated at 28 °C for 48 h. The clear halos observed around bacterial colonies indicating lipase activity were measured.

Proteases activity was tested on skim milk agar medium [53]. Two microliters of the OD_600_ = 0.1 bacterial suspension were spotted on this plate and incubated at 28 °C for 48 h. The protease activity caused degradation of casein in the medium resulting in clear halo zones around the colonies which were measured.

The described experiments were performed in three biological repetitions with four technical repetitions each.

#### 2.4.3. Biochemical Profile

Biochemical features of the IFB5637 isolate were determined with the use of the API 20E (Biomérieux, Craponne, France) commercial assay including the following tests: ONPG (β-galactosidase production), CIT (utilization of citrate), VP (production of acetoin), GEL (gelatinase production), ADH (arginine dihydrolase), LDC (lysine decarboxylase), ODC (ornithine decarboxylase), H_2_S production, URE (urease production), TDA (utilization of tartarate), IND (indole production), VP (acetoin production), GEL (gelatinase production), GLU (glucose fermentation), MAN (mannitol fermentation), INO (inositol fermentation), SOR (sorbitol fermentation), RHA (rhamnose fermentation), SAC (saccharose fermentation), MEL (melibiose fermentation), AMY (amygdalin fermentation), and ARA (arabinose fermentation). The previously prepared OD_600_ = 0.1 bacterial suspension was used for inoculation of the API microtubes prior to 24 h incubation at 28 °C. The results were collected and interpreted according to the provided manufacturer’s instructions.

#### 2.4.4. Other Factors Involved in Virulence

Bacterial ability to secrete siderophores was determined on chrome azurol S-agar plates [54]. Two microliters of the OD_600_ = 0.1 bacterial suspension were spotted on this plate and incubated at 28 °C for 48 h. The orange halo zones that formed around the colonies were measured. This experiment was carried out in three biological repetitions with four technical replications each.

Motility assays were performed on TSA medium solidified either with 0.3% agar or 0.6% agar, regarding swimming or swarming [55], respectively. Two microliters of the OD_600_ = 0.1 bacterial suspension were spotted on these plates and incubated at 28 °C for 24 h. The diameters of the bacterial colonies were measured after 24 h. Each test was carried out in three biological repetitions, each involving four technical ones.

Salt tolerance was assayed on TSA medium with 5% NaCl [56]. Two microliters of the OD_600_ = 0.1 bacterial suspension were spotted on this plate and incubated at 28 °C for 48 h. Salt tolerance was stated if bacterial growth was observed on such a plate.

Bacterial ability to form biofilm was determined as described by Nykyri et al. [57]. In more detail, 10 µL of an overnight bacterial culture in TSB was added to Eppendorf tubes containing 400 µL of M9 minimal medium [58]. After 16 h of static incubation at 18 °C, bacterial biofilm formed on the walls of the tubes. For staining, 70 µL of 1% crystal violet solution was added into each tube and incubated at room temperature for 20 min. The tubes were washed with distilled water and air-dried. Subsequently, 600 µL of 96% ethanol was utilized to extract the absorbed crystal violet dye. One hundred microliters of the resultant suspension were placed in a 96-well microtest plate and quantified spectrophotometrically at 565 nm with a Victor2 I420 Multilabel Counter (Wallac, Finland).

Isolation and determination of the sugar composition of the O-polysaccharide (OPS) of bacterial lipopolysaccharide (LPS) was performed as described by Kowalczyk et al. [59]. The analyzed strain was plated from frozen stock and cultured on TSA medium at 28 °C for 24 h. Then, 200 mL TSB medium was inoculated with a single bacterial colony taken from the TSA plate and subsequently incubated at 28 °C with 130 rpm shaking for 24 h. This overnight culture was used for inoculation of 2 L of TSB medium. Incubation for 48 h at 28 °C with 130 rpm shaking followed. Bacterial cells were harvested using the centrifuge 6-16KS (Sigma, Kanagawa, Japan) at 16,915.34× *g* for 10 min and stored at −20 °C. The above-described procedure was repeated until 50 g of the IFB5637 strain biomass was collected. The LPS was isolated from dried bacterial cells using hot phenol extraction and purified by enzymatic digestion (DNAse, RNAse, and Proteinase K). The OPS was obtained after hydrolysis of the LPS with acetic acid and separation by size exclusion chromatography. Finally, sugar analysis (hydrolysis with trifluoroacetic acid, reduction with sodium borohydride, acetylation in acetic anhydride, and analysis by gas chromatography-mass spectrometry technique) was performed to identify and determine the number of monosaccharide residues in the O-polysaccharide.

### 2.5. Statistical Analysis

Statistical analysis of phenotypic data was conducted using R, version 3.3.2 (31 October 2016) (R Core Team 2014, Vienna, Austria). Levene’s test was applied for testing the equality of variances and the Shapiro–Wilk’s evaluation was implemented for evaluating normality of the results. Depending on the outcomes of the above-listed tests, either ANOVA followed by Tukey’s honest significance test or Kruskal–Wallis test together with a post-hoc analysis applying Fisher’s least significance criterion were utilized. *p* < 0.05 was applied in all the calculations.

## 3. Results and Discussion

The *P. aquaticum* IFB5637 strain was isolated from water samples collected on (2 October 2016) from Jelen Lake from 0 m depth nearby the shore. Jelen Lake is a kettle lake located in the Polish region of Kashubia. According to a 2006 report, this lake was in the 1st class in terms of water purity, i.e., it exhibited a low level of organic matter, minimal numbers of biogenic substances in addition to low concentrations of dissolved inorganic matter [60]. Of the parameters analyzed, only the elevated level of total nitrogen was confirmed [60]. When the water was sampled, the air temperature was 17 °C and the water temperature was 16 °C. As the obtained *P. aquaticum* IFB5637 isolate was the first strain from this species isolated from the waterways in Poland, we performed its detailed characterization.

### 3.1. Genotypic Features of P. aquaticum IFB5637 Strain

The BLAST comparison of the fragments of gene sequences of *dnaX* and *recA* of the identified isolate and the sequences of *Pectobacteriaceae* strains deposited in the GenBank database showed 99.38% and 97.38% identity in addition to 100% and 93% coverage, respectively, between the *dnaX* and *recA* sequences of the acquired pectinolytic isolate *P. aquaticum* IFB5637 and the corresponding sequences of *P. aquaticum* CFBP 8637^T^ (accession numbers TMK516879.1 and MK517167.1) [35]. The sequences of *dnaX* and *recA* genes of *P. aquaticum* IFB5637 were submitted to GenBank and are available under the following accession numbers: MW657238 and MW660584. Regarding single-nucleotide polymorphisms (SNPs) detected in the 535-bp *dnaX* gene fragments of *P. aquaticum* isolates (IFB5637, CFBP 8632, CFBP 8636, CFBP 8633, CFBP 8637^T^, CFBP 8636, CFBP 8634), we were able to distinguish two SNP regions (at 222 bp position and at 451 position) in which the C–T transition occurred in *P. aquaticum* IFB5637. The *recA* gene sequence of IFB5637 was identical to the sequence of CFBP 8636 strain and the other *P. aquaticum* (CFBP 8632, CFBP 8633, CFBP 8637^T^, CFBP 8636, CFBP 8634).

A neighbor-joining tree, constructed on the sequences of the *dnaX* (Figure 1) and *recA* (data not shown) gene fragments of the tested isolate in comparison to the reference *Pectobacteriaceae*. (Table 1) strains, grouped the sequences of the studied isolate and the sequences of the reference strains of *P. aquaticum* into one distinct clade.

The conducted genomic profiling with the use of ERIC primers (Figure 2), revealed small differences between the tested *P. aquaticum* IFB5637 and the reference strains of *P. aquaticum*, i.e., CFBP 8637^T^, CFBP 8632, CFBP 8633, CFBP 8634, CFBP 8635, and CFBP 8636. The acquired ERIC patterns for CFBP 8636, CFBP 8632, and CFBP 8634 strains of *P. aquaticum* isolated in 2015 were less heterogenous than in the case of the other two *P. aquaticum* strains (CFBP 8633, CFBP 8635) from 2016. Similar results were presented in the publication by Zoledowska et al. [11], where differences in the profiles of *P. parmenteri* isolates obtained in Poland in different years were observed. However, the genomic profiles of *P. aquaticum* strains were not as homogeneous as in the case of *Dickeya solani* isolates, as presented in a study by Golanowska et al. [61]. In that research, REP genomic profiles of all tested *D. solani* strains from Poland, Finland, and Israel turned out to be exactly the same as the molecular pattern of *D. solani* IPO 2222^T^.

### 3.2. Phenotypic Features of P. aquaticum IFB5637 Strain

The studied *P. aquaticum* IFB5637 of water origin showed an ability to macerate potato tubers and chicory leaves. In more detail, the analyzed isolate exhibited weaker virulence on potato and chicory in comparison to the two tested *P. brasiliense* (CFBP 6617 and HAFL05) and two *P. carotovorum* (SCRI 136 and IFB5637) strains (Figure 3). However, its ability to macerate plant tissue was higher than that of *P. atrosepticum* SCRI 1086. Moreover, in the previous work [35], *P. aquaticum* strains isolated from water in France were shown to be capable of causing disease symptoms on potato tuber tissue.

The *P. aquaticum* IFB5637 strain turned out to be able to produce a wide spectrum of plant-cell-wall-degrading enzymes (Table 2). It is worth noticing that *P. aquaticum* IFB5637 showed a significantly lower pectinase and protease activities than the tested *P. brasiliense* (CFBP 6617 and HAFL05) and *P. carotovorum* strains (SCRI 136 and IFB5369). The low pectinases and proteases activities exhibited by the analyzed *P. aquaticum* IFB5637 (Table 2) were correlated with the low maceration capacity towards potato and chicory tissues. Similar outcomes were reported by Potrykus et al. [62] during the study of the closely related *D. solani* strains. In that work, the low virulent *D. solani* IFB0223 strain did not show any protease activity and significantly lower pectinase activity, in contrast to the highly virulent *D. solani* IFB0099 strain, which exhibited high pectinase and protease activities. Moreover, the above-listed *D. solani* strains differed in the ability to macerate plant tissue.

Concerning the capacity of *P. aquaticum* IFB5637 to chelate iron ions, we observed a relatively high potency of this strain to produce siderophores (Table 2). On the other hand, the reference strains *P. parmentieri* SCC3193 and *P. atrosepticum* SCRI 1086 revealed no siderophore production ability, which is consistent with the results obtained by Zoledowska et al. [11].

The motility assay indicated that *P. aquaticum* IFB5637 is capable of moving, either by swimming or swarming (Table 2). Interestingly, the *P. aquaticum* IFB5637 strain from water exhibited lower swarming and swimming capacities than the closely related *P. brasiliense* strains: CFBP 6617 and HAFL05 (Table 2). The obtained results are in agreement with the data shown in a study by Ozturk et al. [41], where the lowest swimming motility was presented by the *P. atrosepticum* isolates while the highest by *P. brasiliense* and *P. carotovorum* strains.

Additionally, *P. aquaticum* IFB5637 exhibited a lower ability of biofilm formation (Table 2) than the other tested *Pectobacteriaceae* strains. It is worth noting that one of the tested strains of *Pectobacteriaceae* (*P. brasiliense* CFBP 6617) exhibited biofilm production at the same level as the *Pseudomonas aeruginosa* ATCC 15692 strain included as a positive control in this experiment. Besides, both intra-species variation and inter-species variation in the biofilm formation potency could be observed.

Based on the API 20E assay, it was found that the *P. aquaticum* IFB5637 strain was able to ferment mannitol, rhamnose, saccharose, amygdalin, and arabinose like all the other *Pectobacterium* spp. tested (Table 3). Similar results for the reference strains were described in the study by Ozturk et al. [41], which confirmed the ability of the analyzed strains of *P. atrosepticum*, *P. carotovorum* subsp. *brasiliense*, *P. carotovorum* subsp. *carotovorum*, and *P. parmentieri* to utilize L-rhamnose, D-mannitol, raffinose and L+arabinose. However, contrary to the investigated *Pectobacterium* spp. strains, *P. aquaticum* IFB5637 did not degrade melibiose (Table 3). The *P. aquaticum* IFB5637 strain was also stated to be capable of gelatinase production in contrast to the closely related *P. brasiliense* strains CFBP 6617 and HAFL05. Moreover, all the herein tested *Pectobacterium* spp. strains, including *P. aquaticum* IFB5637, were able to produce β-galactosidase and acetoin. The *P. aquaticum* IFB5637 strain was shown to be unable to utilize sorbitol and to grow on 5% NaCl (Table 3) similar to the *P. aquaticum* strains investigated by Pedron et al. [35].

The analysis of the chemical structure of *P. aquaticum* IFB5637 LPS revealed that the *P. aquaticum* IFB5637 strain produces a smooth form of LPS, which is known as one of the important virulence factors responsible for adherence of bacteria and plant tissue colonization. The OPS of *P. aquaticum* IFB5637 LPS turned out to contain three different monosaccharides, namely, mannose, glucose, and abequose (3,6-dideoxy-D-xylo-hexose). Mannose and glucose are relatively common monosaccharides in bacterial O-polysaccharides, while abequose has not been identified yet in any strains of the *Pectobacteriaceae* family [59,63,64].

As suggested by Pedron et al. [35], divergence within the *Pectobacterium* species outside the plant context is highly understudied. The herein presented data contribute to broadening the knowledge on the newly established *P. aquaticum* species. As far as we are aware, the herein presented research is the second study on genotypic and phenotypic features of this surface-water-associated pectinolytic species. We found that sequencing of *dnaX* [45] and *recA* [46] genes could be used for distinction of *P. aquaticum* species. Moreover, in view of the unavailability of *P. aquaticum*-specific PCR reactions, Pedron et al. [35] observed that the sequencing of *gapA*, in contrast to 16S rRNA, is informative enough for differentiation of *P. aquaticum* strains. In accordance with high dDDH and pairwise ANI values, equaling >86% and ≥98%, respectively, as calculated by Pedron et al. [35], we herein observed rather small differences in the ERIC-based genomic profiles of all the previously described *P. aquaticum* strains (Figure 2). We also reportedthe API 20E biochemical profile of Paq IFB5637; however, it did not discriminate this strain from, for example, *P. atrosepticum* SCRI 1086. Therefore, API 20E profiling is not distinctive enough in contrast to Omnilog-based identification of Pedron et al. [35] involving, for instance, metabolic reactions with trehalose, cellobiose or formic acid that differentiated *P. aquaticum* from the other tested *Pectobacterium* spp. We observed that *P. aquaticum* IFB5637 was able to efficiently macerate potato slices and chicory leaves; this is in accordance with Pedron et al. [35], who reported the maceration capacity of *P. aquaticum* strains on potato tubers under laboratory conditions. Notably, we studied for the first time in *P. aquaticum* the activities of plant-cell-wall-degrading enzymes, which are the most important virulence factors of soft rot *Pectobacteriaceae*. Even if the noted action of cellulases and proteases was lower than, for example, that of highly virulent *P. brasiliense* strains, it was higher than the one attributed to the included *P. atrosepticum* strain, isolated from the rotted potato tubers. *P. aquaticum* IFB5637 also showed significant production of siderophores and high swimming motility which further supports its virulence potential. Importantly, we provided the first insights into the sugar composition of O-polysaccharide of *P. aquaticum* LPS. LPS molecules are associated with the virulence of bacterial strains, especially during colonization and overcoming of the defense mechanisms of the plant host. The herein presented data suggest that the usage of surface waters containing pectinolytic bacteria, even if the members of a certain taxonomic group have not been previously isolated from the affected plant tissue, for the irrigation of potato fields can potentially contribute to the spread of soft rot and blackleg diseases.

## 4. Conclusions

To the best of our knowledge, this is the first report on the isolation of the *P. aquaticum* strain in Poland. *P. aquaticum* IFB5637 acquired from Jelen Lake was capable of macerating potato and chicory tissues. This strain exhibited the activity of plant-cell-wall-degrading enzymes that are important virulence factors. The study on the OPS of the LPS of *P. aquaticum* revealed the presence of common sugars, such as mannose and glucose in addition to abequose, identified for the first time among the *Pectobacteriaceae* family. The presented data suggest that the usage of surface waters for irrigation of potato fields can potentially contribute to soft rot bacteria spread.

## Figures and Tables

**Figure 1 ijerph-18-05041-f001:**
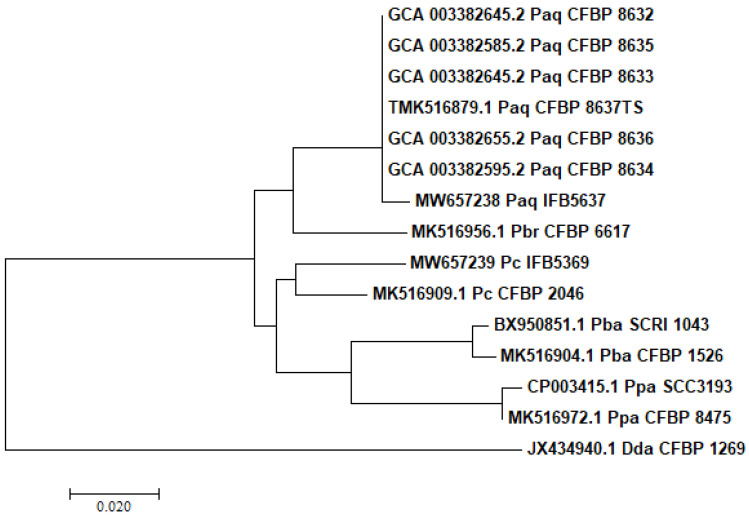
Phylogenetic analysis based on 535 bp fragment of *dnaX* gene. *P. aquaticum* IFB5637 isolated in Poland was juxtaposed to the *Pectobacteriaceae* reference strains. The sequences of the utilized strains: *P. aquaticum* (Paq CFBP 8632, Paq CFBP 8635, Paq CFBP 8633, Paq CFBP 8637^T^, Paq CFBP 8636, Paq CFBP 8634, *P. brasiliense* (Pbr CFBP 6617), *P. carotovorum* (Pc CFBP 2046), *P. atrosepticum* (Pba SCRI 1043, Pba CFBP 1526), *P. parmentieri* (Ppa SCC3193, Ppa CFBP 8475), and *D. dadantii* (Dda CFBP 1269) were downloaded from the GenBank database (07.2020). The tree was generated with a neighbor-joining algorithm with the Jukes–Cantor nucleotide distance measure in the MEGA X software package. Bootstrap was set on the level of 1000 replicates.

**Figure 2 ijerph-18-05041-f002:**
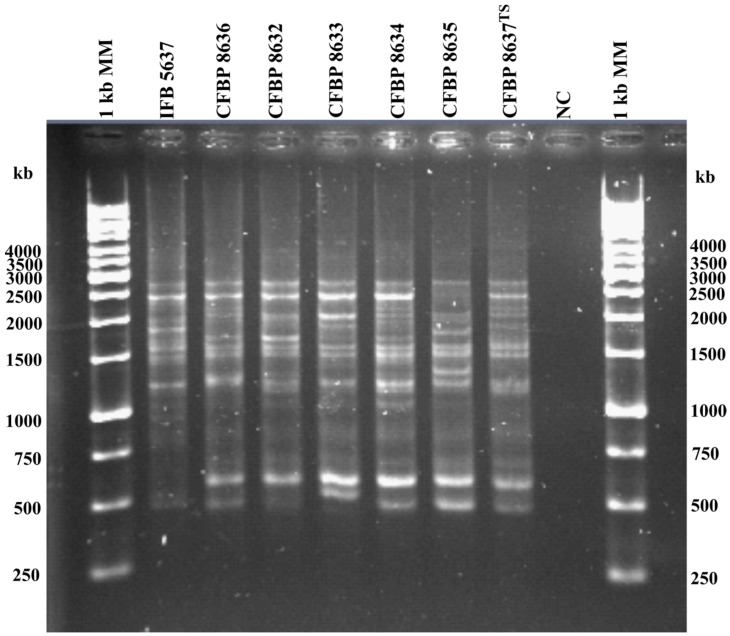
ERIC-based genomic profiling of *P. aquaticum* strains. MM—1-kb DNA Ladder (Thermo Fisher Scientific, Minneapolis, MN, USA); IFB5637—*P. aquaticum* isolated in the frames of this study; *P. aquaticum* reference strains: CFBP 8637^T^, CFBP 8632, CFBP 8633, CFBP 8634, CFBP 8635, and CFBP 8636; NC—negative control.

**Figure 3 ijerph-18-05041-f003:**
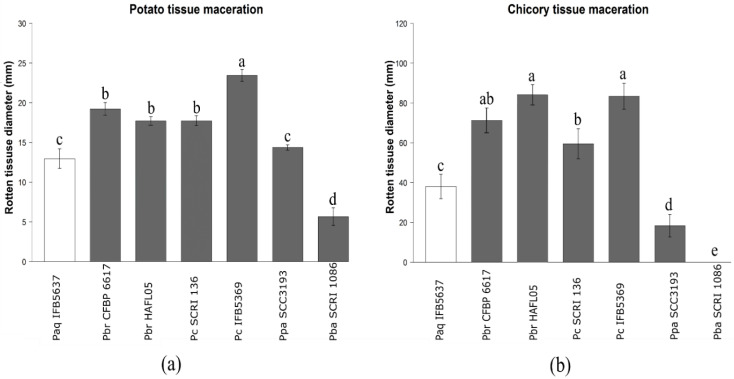
Comparison of the potato (**a**) and chicory (**b**) tissue maceration ability of the tested *P. aquaticum* IFB5637 strain isolated from water in Poland in contrast to the closely related *Pectobacteriaceae* strains. Strains abbreviations: *P. aquaticum* (Paq IFB5637), *P. brasiliense* (Pbr CFBP 6617, Pbr HAFL05), *P. carotovorum* (Pc SCRI 136, Pc IFB5369), *P. parmentieri* (Ppa SCC3193), and *P. atrosepticum* (Pba SCRI 1086). Means ± SE of diameters (potato) or lengths (chicory) of the rotten tissues are depicted. Three independent experiments with nine technical replications were conducted. Means marked with different letter (a, b, c, d, e) are significantly different according to the Kruskal–Wallis test followed by a post-hoc analysis applying Fisher’s least significant criterion at *p* < 0.05.

**Table 1 ijerph-18-05041-t001:** Strains of pectinolytic bacteria used in this study.

Species	Strain ^a^	Source, Country of Isolation	Year of Isolation	Reference
*Pectobacterium atrosepticum*	SCRI 1086 IFB5103	*Solanum tuberosum*, Canada	1985	[39]
*Pectobacterium carotovorum*	SCRI 136 IFB5118	*Solanum tuberosum*, USA	NA	[40]
	IFB5369	*Solanum tuberosum*, Poland	2011	[41]
*Pectobacterium brasiliense*	CFBP 6617 LMG 21371 IFB5390	*Solanum tuberosum,* Brazil	1999	[16]
	HAFL05 IFB5527	*Solanum tuberosum,* Switzerland	2013	[42]
*Pectobacterium parmentieri*	SCC3193 IFB5308	*Solanum tuberosum*, Finland	1980s	[43]
*Pseudomonas aeruginosa*	ATCC 15692 PAO1 IFB9036	infected wound, Australia	1955	[44]
*Pectobacterium aquaticum*	CFBP 8637^T^ NCPPB 4640 A212-S19-A16	fresh water, France	2016	[35]
	CFBP 8636 A127-S21-F16		2015	
	CFBP 8632 A35-S23-M15		2015	
	CFBP 8633 A101-S19-F16		2016	
	CFBP 8634 A104-S21-F16		2015	
	CFBP 8635 A105-S21-F16		2016	
	IFB5637	fresh water, Poland	2016	this study

NA—not available. ^a^ IFB—Intercollegiate Faculty of Biotechnology University of Gdansk and Medical University of Gdansk, Poland; SCRI—The James Hutton Institute, bacterial collection, Scotland; CFBP—Plant–Associated Bacteria Collection, France; SCC—World Federation for Culture Collections, Netherlands; ATCC—American Type Culture Collection, USA; HAFL05—School of Agricultural, Forest, and Food Sciences HAFL, Switzerland; LMG—Belgian Coordinated Collections of Microorganisms (BCCM), Belgium; NCPPB—National Collection of Plant Pathogenic Bacteria, France.

**Table 2 ijerph-18-05041-t002:** Phenotypic characteristics of *P. aquaticum* IFB5637 strain isolated in Poland in comparison to the tested *Pectobacteriaceae* strains.

Activities	*P. aquaticum* IFB5637	*P. brasiliense* CFBP 6617	*P. brasiliense* HAFL05	*P. carotovorum* SCRI 136	*P. carotovorum* IFB5369	*P. parmentieri* SCC3193	*P. atrosepticum* SCRI 1086
Pectinases activity	11.0 ± 2.7	19.3 ± 1.2	22.0 ± 1.5	22.0 ± 1.7	22.3 ± 0.8	16.9 ± 1.7	10.8 ± 0.7
Proteases activity	6.3 ± 1.4	14.3 ± 1.5	16.4 ± 1.4	13.0 ± 1.4	19.7 ± 1.3	5.7 ± 1.2	2.9 ± 1.5
Cellulases activity	13.5 ± 0.8	6.0 ± 1.4	12.1 ± 0.7	11.7 ± 0.8	13.6 ± 0.9	0.0 ± 0.0	0.0 ± 0.0
Lipases activity	13.4 ± 0.5	16.8 ± 0.5	19.3 ± 0.6	17.5 ± 0.6	13.5 ± 0.9	15.3 ± 0.4	9.3 ± 0.5
Siderophores activity	7.1 ± 0.2	10.8 ± 0.3	2.8 ± 1.2	4.5 ± 1.0	7.8 ± 0.7	0.0 ± 0.0	0.0 ± 0.0
Swimming	22.8 ± 1.7	39.8 ± 3.2	41.6 ± 1.5	34.2 ± 1.6	39.4 ± 2.2	13.0 ± 0.7	5.2 ± 0.4
Swarming	4.0 ± 0.4	10.5 ± 4.1	9.3 ± 3.5	4.8 ± 0.7	14.4 ± 5.6	9.7 ± 3.6	4.1 ± 0.2
Biofilm formation *	+	+++	++	+	+	++	+

* The biofilm formation capacity of *P. aquaticum* IFB5637 was compared to that of *Pseudomonas aeruginosa* ATCC 15692 strain. The collected results are designated as follows: +++ strain was able to form biofilm as efficiently as *P. aeruginosa* ATCC 15,692; ++ strain was capable of forming biofilm significantly less efficiently than the reference strain; + marks the least potent biofilm producers.

**Table 3 ijerph-18-05041-t003:** Biochemical features of the *P. aquaticum* IFB5637 strain in comparison to the other tested *Pectobacteriaceae* strains.

Biochemical Features	*P. aquaticum* IFB5637 ^T^	*P. brasiliense* CFBP 6617	*P. brasiliense* HAFL05	*P. carotovorum* SCRI 136	*P. carotovorum* IFB5369	*P. parmentieri* SCC3193	*P. atrosepticum* SCRI 1086
ß-galactosidase production	+	+	+	+	+	+	+
Arginine dihydrolase production	−	+	−	−	+	−	−
Lysine decarboxylase production	−	−	−	−	−	−	−
Ornithine decarboxylase production	−	−	−	−	−	−	−
Utilization of citrate	+	+	+	+	+	−	+
H_2_S production	−	−	−	−	−	−	−
Urease production	−	−	−	−	−	−	−
Utilization of tartrate	+	+	+	+	+	+	+
Indole production	−	−	+	−	−	−	−
Production of acetoin	+	+	+	+	+	+	+
Gelatinase production	−	+	+	−	+	−	−
Glucose fermentation	−	−	−	−	−	−	−
Mannitol fermentation	+	+	+	+	+	+	+
Inositol fermentation	−	−	−	−	−	−	−
Sorbitol fermentation	−	−	−	−	−	−	−
Rhamnose fermentation	+	+	+	+	+	+	+
Saccharose fermentation	+	+	+	+	+	+	+
Melibiose fermentation	−	+	+	+	+	+	−
Amygdalin fermentation	+	+	+	+	+	+	+
Arabinose fermentation	+	+	+	+	+	+	+
Growth in 5% NaCl	−	+	+	−	+	−	−

+ Refers to possession of a specific metabolic feature, − lack of such a trait.

## Data Availability

Determined sequences data of *dnaX* and *recA* genes of *Pectobacterium aquaticum* IFB5637 were deposited in GenBank under the following accession number MW657238 (*dnaX* gene) and MW660584 (*recA* gene).

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
