# Peer review of "The First Polish Isolate of a Novel Species Pectobacterium aquaticum Originates from a Pomeranian Lake"

_ijerph, 2021, doi:10.3390/ijerph18095041_

Round 1
Reviewer 1 Report
The Authors isolated and chracterized the first strain of Pectobacterium aquaticum from Poland, which might cause serious troubles in future agriculture. The topic is highly relevant and could generate attention from researchers and farmers alike. The MS is well written and the experiments were carefully designed which lead to interesting results. I congratulate the Authors for this nice work and wish them further success. I would have just two minor comments before its publication.
Page 5, Line 199: Fermentas, USA – that company was a Lithuanian one, to the best of my knowledge but is now part of ThermoFisher Scientific, which is indeed an American firm
Page 9, Line 369: diminished virulence – maybe weaker or lesser would be a more appropriate word
Author Response
Dear Reviewer,
We want to thank you for the review and all comments concerning the study presented in our manuscript IJERPH – 1187517 entitled " The first Polish isolate of a novel species Pectobacterium aquaticum originates from a Pomeranian lake”.
The revision has been performed according to your suggestions. Please find below the detailed answers to you comments.
Also all references were checked and corrected.
English has been checked and corrected by specialist from the University of Gdańsk Foreign Language Department
Answers to the Reviewer
Page 5 Line 199 (currently page 3 line 12; page 5 line 215; page 10 line 393
Thank you for correction; I have changed the name of company from Fermentas to Thermo Fisher Scientific.
Page 9, line 369 (currently page 10. Line 400)
I have changed “diminished virulence” to the “weaker virulence” as suggested.
Sincerely Yours,
Ewa Lojkowska
Reviewer 2 Report
Review Report (ijerph1187517): The first Polish isolate of a novel species Pectobacterium aquaticum originates from a Pomeranian lake.
Abstract: Minor Changes only
Line: 14-17: Make two simple sentences. More than 40 words in a single sentence. This will make text easy to understand.
Line 19: ‘Type Strain’: T and S in uppercase?
Line 19: Expand CFBP (carotovorum subsp. odoriferum). This would help in search for this article in search engines.
Line 19: Expand API-20 (Analytical Profile Index)
Line 21: Did you give this new name (P. aquaticum IFB5637)? Mention it in abstract and also expand IFB.
Line: 22: Cellulose or cellulase? Introduction: Introduction is concise and explains the need of this work. Please check the use of commas at several places in the text.
Line: 44: Do not use ‘they’ for bacteria.
Line 64-65: insects, and ….
Line 66: excessive irrigation and………
Line: 68: Please replace ‘a regarded as’ a potential ………. Material and Method: All the methods are explained in detail.
Line: 94: ‘in addition to’ replace it by ‘and;.
Line: 95: ‘Also one slime sample from 20 m depth was acquired’ replace it by ‘In addition, one slime sample was also acquired at a death of 20 m’.
Line 96: ‘mixing the’ replace by ‘mixing of’
Line 110: ‘bacterial cell lysate’. How bacterial cells were lysed? Explain.
Result and Discussion:
Line 317: Reference is required to support the given statements about lake water quality.
Line: 359: Expand D. solani. 'Dickeya solani' Comments: Authors meticulously planned the experiment and used standard methodology. This work is novel and provided applied information to local potato growers to make timely decisions to protect crops. I recommend acceptance after moderate revision. Good work.
Author Response
Dear Reviewer,
We want to thank you for the review and all comments concerning the study presented in our manuscript IJERPH – 1187517 entitled " The first Polish isolate of a novel species Pectobacterium aquaticum originates from a Pomeranian lake”.
The revision has been performed according to your suggestions. Please find below the detailed answers to you comments.
Also all references were checked and corrected.
English has been checked and corrected by specialist from the University of Gdańsk Foreign Language Department.
Answers to the Reviewer
Page 1 Line 14-17 The sentence:
During 2-years monitoring of the prevalence of soft rot Pectobacteriaceae in different depths of Pomeranian lakes, performed by a qualified scuba diver, we identified and broadly characterized the first isolate in Poland of a newly established species, namely Pectobacterium aquaticum.
Is change to two shorter sentence
The prevalence of soft rot Pectobacteriaceae in different depths of Pomeranian lakes was performed by a qualified scuba diver during 2-years of monitoring. It allowed for the isolation and broad characterization of a strain from the newly established species Pectobacterium aquaticum.
Abstract
Line 14-17 (currently 24) Type strain is corrected to CFBP 8637T
Line 19 (currently 24) CFBP is abbreviated by Collection Francaise de Bacteries Phytopathogenes which in English is Plant-Associated Bacteria Collection, France;
Line 19 (currently 25) 25 API 20E is expanded to Analytical profile index
Line 21 (currently 28) The sentence is changed and information that we have gave this name to isolated strain is given; IFB is a name of our bacteria collection Intercollegiate Faculty of Biotechnology.
Line 22 (currently 29) celluloses is corrected to cellulases.
Use of commas was checked and corrected.
Introduction
Line 44, Line 64-65 , Line 66, Line 68
The sentences are corrected
Line 94 (currently 103)
Sentence is corrected to “In addition, one slime sample was also acquired at a depth of 20 m.”
Page 96 (currently 10)
“Mixing the” is replaced by “mixing of”
Line 110 (currently 119)
The sentence describing how bacterial lysate were prepared is added
The additional sentence was added. After 24 h incubation at 28°C, a single bacterial colony was suspended in 200 µl of sterile distilled water and frozen for 45 min at -20°C and then defrosted. This procedure allows for effective lyase of the bacterial cells.
Results and Discussion
Line 317 (currently 347)
The references is added; Cieśliński 2009.
Line 359 (currently 392)
D. solani is expanded to Dickeya solani
Sincerely Yours,
Ewa Lojkowska
Reviewer 3 Report
The paper titled "The first Polish Isolate of a Novel Species Pectobacterium 2 aquaticum Originates from a Pomeranian Lake" describes the isolation and identification of several Pectobacterium aquaticum isolated from freshwaters in Poland. They identified the strains (sequencing of recA and dnaX and rep-PCR), performed biochemical analysis (API), measured different enzymatic activities and performed a pathogenicity test two crops (potatoe and chicory). Overall the paper is fundamentally descriptive of the strain isolated. The paper is well written and stastically sound.
Minor comments:
Line 68:change “be a regarded as” to “be regarded as”
Line 116: Explain why or/and what is amplified with the different sets of primers used.
Please review the reference list, especially, 43 and 63
Figure 1. Could be improved by including species names, however this is an editorial decision.
Although the authors state why they use recA and dnaX insted of 16SRDNA, I cannot find any seuqneces in NCBI of the recA or dnaX for the strain.
Even though the editorial might not ask for the sequences to be deposited, this would greatly help to achieve the goal of open science.
Author Response
Dear Reviewer,
We want to thank you for the review and all comments concerning the study presented in our manuscript IJERPH – 1187517 entitled " The first Polish isolate of a novel species Pectobacterium aquaticum originates from a Pomeranian lake”.
The revision has been performed according to your suggestions. Please find below the detailed answers to you comments.
Also all references were checked and corrected.
English has been checked and corrected by specialist from the University of Gdańsk Foreign Language Department
Line 68 (currently 76)
“be a regarded as source” is changed to “be a potential sources”
Line 116 (currently 122-128)
Line 116: Explain why or/and what is amplified with the different sets of primers used.
The text of this paragraph is supplemented with the information required by the reviewer.
“This assignment was conducted by a multiplex PCR assay [38]. This reaction was carried out in 25 μl PCR mixture containing 2 μl of the tested bacterial lysate, 1 x reaction buffer supplemented with KCl (Thermo Fisher Scientific, USA), 2.5 µM MgCl2, 80 μM of each dNTPs, 0.32 μM Df and Dr primers (for Dickeya genus), 0.1 μM Y45 and Y46 primers (for P. atrosepticum), 1.2 μM ExpccF and ExpccR primers (for bacteria formerly classified as P. carotovorum) in addition to 1 U of the Taq recombinant DNA Polymerase (Thermo Fisher Scientific, USA).”
References list
The list of references has been checked and corrected
Figure 1.
Could be improved by including species names, however this is an editorial decision.
I have added abbreviation of the species names in the Figure 1 and Figure 3 legends.
Although the authors state why they use recA and dnaX insted of 16SRDNA, I cannot find any seuqneces in NCBI of the recA or dnaX for the strain. Even though the editorial might not ask for the sequences to be deposited, this would greatly help to achieve the goal of open science.
The sequences of recA and dnaX are deposited in Gene Bank under the accession numbers accession numbers MW657238 (dnaX gene) and MW660584 (recA gene). The sequences were not available publicly but today I have asked Gene Bank to make these sequences available for the scientific community.
Sincerely Yours,
Ewa Lojkowska